# Health-Seeking Behaviors in Mozambique: A Mini-Study of Ethnonursing

**DOI:** 10.3390/ijerph19042462

**Published:** 2022-02-21

**Authors:** Naoko Takeyama, Basilua Andre Muzembo, Yasmin Jahan, Michiko Moriyama

**Affiliations:** 1Graduate School of Biomedical and Health Sciences, Hiroshima University, Hiroshima 734-8553, Japan; naoko.tkym@gmail.com (N.T.); dr.yasminjahan@gmail.com (Y.J.); 2Graduate School of Medicine, Dentistry and Pharmaceutical Sciences, Okayama University, Okayama 700-8530, Japan; andersonbasilua@yahoo.fr

**Keywords:** cultural influence, health-seeking behaviors, ethnonursing, Mozambique

## Abstract

In settings where traditional medicine is a crucial part of the healthcare system, providing culturally competent healthcare services is vital to improving patient satisfaction and health outcomes. Therefore, this study sought to gain insight into how cultural beliefs influence health-seeking behaviors (HSBs) among Mozambicans. Participant observation and in-depth interviews (IDIs) were undertaken using the ethnonursing method to investigate beliefs and views that Mozambicans (living in Pemba City) often take into account to meet their health needs. Data were analyzed in accordance with Leininger’s ethnonursing guidelines. Twenty-seven IDIs were carried out with 12 informants from the Makonde and Makuwa tribes. The choice of health service was influenced by perceptions of health and illness through a spiritual lens, belief in supernatural forces, dissatisfaction with and dislike of the public medical system on grounds of having received poor-quality treatment, perceived poor communication skills of health professionals, and trust in the indigenous medical system. This study confirmed the need for health professionals to carefully take cultural influences into consideration when providing care for their patients. We recommend an educational intervention that emphasizes communication skills training for healthcare workers to ensure successful physician/nurse–patient relationships.

## 1. Introduction

Despite significant advances in medical care, in Mozambique, many people still die from diseases for which effective treatments have been established. For instance, ailments such as diarrhea, pneumonia, malaria, and HIV/AIDS fall into this category [1,2]. Inappropriate healthcare-seeking behavior is one of the factors contributing to poor health outcomes, including death, in people suffering from malaria [3] and HIV/AIDS [4].

Mozambique has government healthcare services mainly based on modern medicine that are viewed as the norm and allows, in theory, all citizens to access these services. However, in practice, for most populations, some barriers remain regarding access to formal health services. Factors such as fear of mistreatment by and distrust of care providers, wariness of governments, financial constraints, and ethno-pathogenic perspectives (produced from indigenous causes, not from western medicine) of illnesses have been reported to discourage people from using conventional health services in Mozambique [5]. In addition, epidemiological data support the view that poor accessibility to health facilities and lack of adequate medical infrastructure limit access to conventional health services [6,7,8]. In these circumstances, traditional healers are often the only form of healthcare for many people, especially those living in rural areas, to meet their healthcare needs or understand the causes of their social problems [9]. However, even when conventional health services are available and accessible, some people in Africa still show a preference for traditional healers to deal with illnesses and ailments that they believe to be caused by sources such as spirits and sorcery. About 80% of African citizens, including Mozambicans, rely on traditional medicine [10,11]. The perceived failure of medical treatment, easy accessibility, low cost, familiarity, and respect for culture and dignity have also been reported as factors contributing to the preference for traditional medicine among Africans [12]. Accordingly, if public health strategies are to improve access to conventional health services, they should be based on the cultural aspects that influence health-seeking behaviors (HSBs) [13]. It is also indicated that considering the “spiritual dimension” of health is crucial for introducing a new primary healthcare system in the community [14].

Traditional medicine (TM) has existed in most African communities for hundreds of years because modern medicine was once not readily available [15,16]. Notably, many Africans believe that traditional healers can cure various ailments such as HIV/AIDS. In a study by Audet et al. that involved Mozambican people with HIV/AIDS who were on antiretroviral therapy, the majority of participants relied on traditional healers for their treatment [17]. Because the culture influences the health outcome with effects ranging from positive to negative, it is necessary to understand from a cultural perspective why some people do not choose the treatment regimens established by modern medicine.

Therefore, this study aimed to gain a deeper insight into the way cultural beliefs influence HSBs among Mozambican residents. This is a critical public health goal because health practitioners are not likely to provide culturally sensitive care without understanding the patient’s cultural background [18]. The findings of this study have the potential to contribute towards improved healthcare outcomes among Mozambicans by providing culturally competent care.

### 1.1. Overview of Mozambique

#### 1.1.1. Brief Ethnohistory

Facing the Indian Ocean, Mozambique is situated in southeastern Africa. In 2016, approximately 29 million people inhabited the country. After Vasco da Gama’s arrival in 1498, Mozambique fell under the control of Portugal [19]. After 400 years of foreign governance, the native inhabitants aspired for their independence and achieved their freedom by attacking the Portuguese administrators [20]. Despite attaining independence in 1975, secession from colonial occupation led to a civil war that greatly damaged reconstruction efforts following the war for independence [21].

However, the main ethnic groups in Mozambique are the Makua, Tsonga, Makonde, Shangaan, Shona, Sena, Ndau, and other indigenous groups. There are approximately 45,000 Europeans, and 15,000 South Asians, constituting less than 2% of the population, as well [22]. The literacy rate in Mozambique is 60.7% (male: 72.6%, female: 50.3%) [23], and more than 46% of people were living below the poverty line (USD 0.31 per day) in 2018 [24]. Regarding the religious aspect, 26.2% of citizens are Roman Catholic, 18.3% Muslim, 15.1% Zionist Christian, 27% other religions, and the remaining 13.4% did not list a religious affiliation [25]. Mozambique consists of more than 10 tribes. The official language is Portuguese. Religions and languages vary by region and tribe [26]. The health sector in Mozambique has changed throughout history. Before independence, medical facilities only served the Portuguese officers and those who lived in the capital city [27]. After independence and the civil war, the government achieved the reconstruction of the national health system. Now citizens have free access to public medical facilities under the primary healthcare system. However, access to better private facilities is limited to economically wealthy people and foreigners [27].

#### 1.1.2. Theoretical Guide and Conceptual Framework

Madelaine Leininger asserted the link between providing care and culture in her book, *Culture Care Diversity and Universality* (CCDU), in which she explained that culture and society influence care patterns, expression and practice through environmental context, language and ethnohistory [18]. Therefore, if healthcare professionals understand the cultural influence and experience on HSBs, they can support people by helping them select an effective treatment and improve health outcomes in Mozambique (Figure 1: Conceptual framework). Accordingly, two research questions were addressed in this study: (i) What are the culturally influenced HSBs among Mozambicans living in Pemba City? and (ii) What cultural determinants influence their HSBs the most?

Most past studies on HSBs conducted in Africa have focused on specific diseases such as AIDS/HIV with covariates categorized only according to socio-economic status [28]. Focusing on specific diseases alone, without considering cultural variables, might prevent the integration of general health concepts and HSBs from sociocultural perspectives. The general HSBs of communities in Mozambique have not been sufficiently investigated. Therefore, the researchers conducted this study in Pemba City, Mozambique, without specifying any particular disease.

## 2. Materials and Methods

### 2.1. Study Design

We carried out a mini-ethnonursing study using participant observation (PO) and in-depth interviews (IDIs). This approach was deemed appropriate to gain a deeper understanding of the influence of participants’ cultural beliefs and values on the use of healthcare services. Indeed, using the ethnonursing methodology at all stages of this study helped us gain a grasp of participants’ cultural health beliefs and their decisions about treatments or methods of coping with health problems [29].

### 2.2. Study Area

This study was conducted in Pemba City, which is the capital of Cabo Delgado province situated in northern Mozambique. This study site was chosen considering the principal researcher’s 2-year experience working in a training center with medical professionals in this city. In 2011, Pemba City had one provincial hospital, nine health centers and one health post. It is important to note that Mozambique faces a shortage of healthcare workers. For example, the density of the healthcare workforce is 0.055 doctors and 0.401 nurses and midwives per 1000 people [30]. On the other hand, the density of doctors in neighboring countries (Tanzania, Malawi, Zambia, Zimbabwe, Eswatini and South Africa) is 0.448 per 1000 population, and 1.987 nurses and midwives per 1000 population [31].

### 2.3. Participants and Data Collection

Data collection involved 3 months of PO in medical facilities and the community, and 27 IDIs with 12 informants. The first three key informants were set to obtain credible and rich data from local residents. Key informants were selected based on their knowledge of the local culture and customs as acquaintances of the principal researcher. Two key informants were elementary school teachers. They were also students enrolled in bachelor courses in public administration and ethics. Another key informant was working as a public official with a degree in public administration (Figure 2: Flowchart for data collection process).

### 2.4. Participant Observation

PO was undertaken under natural living conditions, and while staying at the home of the principal researcher’s acquaintance. The principal researcher visited neighbors and colleagues of the host family to observe and explore actions they usually take to resolve health problems. The “stranger to friend” model [18] was used to evaluate the principal researcher’s position in the community. Ultimately, residents and informants trusted the principal researcher. They explained their beliefs and lifestyles, and provided some critical comments on governmental medical facilities.

### 2.5. Recruitment and In-Depth Interviews

We used a theoretical sampling method [32] for recruitment. Sampling was guided by research questions and by stories obtained during PO. Recruitment was initiated with persons who had malaria, which was observed frequently in PO. Then, recruitment was expanded to other informant categories regardless of gender, education level, occupation, or type of health problem. Finally, an administrator and a midwife from a provincial hospital were enlisted to gain health professionals’ views on the current situation with the medical facilities.

The inclusion criterion for recruitment was being a resident of Pemba City. Temporary residents were excluded. The IDI schedule and interview locations were set by the informants and their privacy was strictly maintained. The IDIs were performed using a semi-structured interview guide (Table 1) and conducted by the principal researcher in Portuguese. The interview guide was developed on the basis of the PO. One key informant who spoke all local languages in Pemba City also participated in the IDIs as an interpreter when requested and allowed by the informants. Twelve people participated in the IDIs at least twice to ascertain the credibility of the responses. Each IDI lasted about 30 to 90 min. After collecting socio-demographic data, the IDIs began with the following question: “Please talk about your perception of health” and “what action do you take when having health problems?” Although an interview guide was used, informants could talk freely to provide as much information as possible on HSBs. Detailed notes were taken during the IDIs, and the principal researcher summarized the crucial observations in field notes. IDI contents were recorded on a digital voice recorder after obtaining the informants’ permission. Furthermore, in order to understand the situation holistically, data were gathered in various ways such as listening to local radio and shopping at local markets.

### 2.6. Data Analysis

The researchers analyzed the data qualitatively in accordance with Leininger’s ethnonursing guidelines [18]. First, all collected raw data were documented and coded. Second, the coded data were compared for similarities and differences, then classified according to the domain of inquiry and research questions. Third, the researchers scrutinized all information to discover recurring patterns. In addition, researchers confirmed the saturation of ideas. The Sunrise enabler model [18] helped the researchers to understand the nature of patterns in cultural contexts. Fourth, patterns were interpreted and grouped according to their contextual meaning. Then, major themes were found and formulated in accordance with CCDU. We employed the qualitative criteria of CCDU [18] to ensure that the collected data were reliable.

## 3. Results

### 3.1. Socio-Demographic Characteristics of Informants

Informants (*n* = 12) were from the two main tribes (Makonde and Makuwa) of Pemba city. They ranged in age from their 20s to their 50s, and were from diverse occupational and social backgrounds (Table 2).

### 3.2. Themes Discerned from Patterns

We identified four themes (perceptions of health and illness, local perception of causes of diseases, trust in the indigenous medical system, and dissatisfaction with governmental medical services) with twelve categories and 30 codes (Table 3) through an analysis of all data. These themes are described as follows.

Theme I: Perceptions of health and illness. Participants often viewed health and their experiences with illness through a spiritual lens. They reported being active in seeking information about health problems and expressed a strong desire to take various measures to protect their health. For example, one informant who had suffered from diabetes shared: “I don’t wait without doing anything. I try to look for effective treatment until I find that”.

They believe that their health means free from pain or discomfort from illness. Health and illness are binary oppositions. Contrarily, health is described as “feeling comfortable, eating well and passing stool and urine”. Participants expressed that they needed to be in good health to work effectively and serve their families. One informant explained, “When I am in good health, I feel that my body feels good.... no complaints at all”. Another informant stated that “… without good health, life and family would also be affected negatively”.

Theme II: Local perception of causes of diseases. We derived theme II from the following statements. One key informant said, “For us Africans, there are two causes of diseases. One is attributed to natural factors, and others are traditional diseases by magic”. Another key informant commented, “Truly there is an African Magic here”. Additionally, he said, “The latter is not for hospital. It has to be treated traditionally with magic.”

We interpreted these quotes as implying that Pemba’s residents clearly perceived that evil spirits and supernatural forces were the causes of illness. They believed that there were two types of disease: one type that occurred spontaneously, and another caused by magic. We confirmed this thought with all IDIs and PO data regardless of gender, age, and educational level. Magic is believed to be due to an African sorcerer who causes physical and mental diseases such as schizophrenia.

Theme III: Trust in the indigenous medical system. Theme III is supported by the following quotes: “When people are suffering with Magic, they have to go to a traditional healer. They should not think of going to hospitals”. With this quote, Pemba’s residents clearly divided healthcare systems into two types. One was modern, governmental, and imported from Europe. The other was traditional, which was an integration of medicinal herbs, divination, and Magic (Table 4).

When people explained the healthcare systems, they mostly talked about TM. TM was called curandeirismo and its healers were called curandeiros. Public medical facilities were also recognized to treat health problems. However, TM was more highly respected and trusted among people irrespective of educational level and social status. People traveled far to seek out curandeiros and would pay them large amounts of money.

This behavior was repeatedly confirmed in PO and IDIs. Therefore, TM was viewed as a culture that people were proud of and inseparable from their regrets about colonial history. For example, one informant said, “When Portuguese attained colonization, our cultures and knowledge of certain medicinal plants began to disappear. However, it had been certainly valid and believed. Our culture was swapped for Portuguese culture”.

Theme IV: Dissatisfaction with governmental medical services. Informants were concerned about the unfriendly environment in hospitals among patients and health professionals. They sometimes reported that healthcare professionals repeated only an ambiguous explanation. Residents were disappointed with modern medicine, because it did not clarify the causes of ill health. They expressed their frustration with the insufficient care from biomedical practitioners. They felt that health professionals explained only symptoms such as headaches, and their poor communication skills reduced people’s trust in medical facilities’ treatment. Some participants even felt that encounters with some health professionals could put their health at risk. On the other hand, treatment by curandeiros was considered the best way to solve health problems. For example, one informant commented on curandeiros’ treatment, *“Curandeiros’ exam finds out what caused the disease and who did it…they will bring the medicine to overthrow that Magician or seek medicine to treat natural (pathological) diseases immediately”*.

Curandeiros precisely corresponded to people’s needs, and people were assured of satisfaction with TM. Thus, TM attracted people, as can be seen in the following comments: *“Then I lost confidence in the hospital. I don’t think coming back to the hospital anymore. I select an alternative as a curandeirismo even it is costly I will go back to the Curandeiros”.*

## 4. Discussion

This mini ethnonursing study sought to understand the role of culture in HSBs among Mozambicans living in Pemba City. Overall, findings showed that people’s local perceptions reflect a belief in supernatural forces, something that is called “magic”, and supported TM as an established healthcare system; i.e., TM was the most popular first line of treatment sought by most people who were explored in this study. Similar observations have been made in earlier studies in developing countries [15,33] and they concur with the views of medical anthropologists reporting that medicine reflects the values of those who use it [34].

Research findings also revealed that the cultural beliefs and practices of Pemba citizens affected their HSBs. For instance, Pemba citizens added cultural aspects to health issues using the word “curandeirismo” in an environment that evoked strong feelings of revulsion towards Mozambique’s colonial history. Curandeiros were always close to the people, and their warm attitude earned people’s trust and satisfaction. Visiting curandeiros might be the first step in solving people’s health problems. People chose treatment from faraway traditional healers rather than nearby medical facilities. This indicated people trusted their traditional medicine more than modern medicine, which had been brought by colonial rule. They recognized that modern medicine would be an effective solution to health problems. However, they still preferred TM due to their dissatisfaction with modern medicine, including the poor interpersonal communication skills of health professionals. People applied their beliefs and views regarding the causes of diseases to HSBs that were more likely to help them meet their health needs. People have been looking for more effective treatments by navigating between modern medicine and TM. We surmised that the use of TM on an equal footing with modern medicine by this population could be addressed by encouraging collaborative workshops between traditional healers and modern health providers.

The World Health Organization (WHO) recognized the importance of working with religious actors in the spiritual domain to deal with primary healthcare for changing the lifestyles and health behavior of community people [14]. Medical professionals also need to be aware of the need to collaborate with traditional healers [35]. Indeed, an educational intervention with traditional healers could promote referrals of patients to hospitals [17]. Therefore, this study suggests the need for collaboration between the two medical systems and managing the quality of TM as a health policy, in order to provide people with adequate healthcare.

Under-equipped medical facilities and health professionals’ cold attitudes and rudeness towards patients are already recognized as barriers to healthcare services [5]. As also shown in several studies [15,33,36], Pemba residents demonstrated a preference for TM to meet their health needs. Likewise, this study revealed that TM was more than just an alternative to compensate for a scarcity of medical resources. TM was often seen as a proud culture that has survived its legacy of colonization and is considered as an effective form of indigenous knowledge to deal with health problems. That is why it was often selected as the first choice among available treatment options.

African countries are multiracial, and, therefore, contain quite diverse socio-cultural backgrounds [37]. In this context, to support people in terms of their expectations regarding healthcare choices, we must not fail to give sensitive consideration to cultural aspects, especially if the healthcare provider and the recipient are from different tribes. As health-care providers are not obliged to comply with all patient requests, a lack of communication and understanding of the patient’s culture by healthcare providers may not be helpful in improving health outcomes of people living in Pemba City. Leininger asserted that knowledge of cultural diversity was essential for nurses to provide appropriate care to people.

Even though Pemba City residents have recognized that medications based on modern medical treatment are likely to be the norm and effective, some have not yet realized it in practice due to conflicting relationships with health professionals. Although traditional healers are not likely to have any medical training, many communities in Africa rely on their healing practices to meet their primary healthcare needs [15,16]. However, without national regulations on TM, this reliance on TM may sometimes worsen diseases for which there is an established treatment regimen based on modern medicine. As a result, if people do not use modern healthcare, modern medicine cannot demonstrate its effectiveness, and people are likely to become even more disappointed. This is a vicious circle between modern medicine and TM, which cuts and breaks the link between the two healthcare systems.

TM’s healers gain people’s trust through their communication skills and openness to clients’ cultural backgrounds [38,39]. Such good relationships with people strengthen their recognition; for example, some people stated “we are Africans, we have our own medicine that meets our needs”. People’s cultural backgrounds vary as well as their social backgrounds and educational levels. Therefore, nurses and other healthcare professionals need advanced communication skills to serve patients from diverse cultural backgrounds. However, it is difficult for individuals to acquire such skills. The training of health professionals in Mozambique, and in Pemba City in particular, should emphasize the importance of understanding cultural diversity and universality. In addition, it is necessary to strengthen or equip hospitals with the necessary medical equipment and supplies to meet the expectations of people. We surmise that implementing all of these suggestions could help reduce the harm caused by treatable diseases.

The findings of this study show that it can be difficult to achieve extensive health coverage through a simple strategy such as improving access to medical facilities.

### Limitations of This Study

The first limitation of this study was the language barrier, because Portuguese was not the first language of the researchers or informants. The interpreter, however, was very familiar with local customs, and he was fluent in all local languages spoken in Pemba city. Additionally, translation between Portuguese and English was conducted meticulously to avoid losing any meaning from the original statements. The second limitation was the transferability of the results. The ethnic structure is diverse in the sub-Saharan region, meaning that a country in this region could have diverse cultural backgrounds. Thus, this study’s findings may require special consideration for local culture and ethnic diversity. The third limitation was the short-term PO. The main researcher, however, lived in the research field for 2 years, and had visited Pemba City many times since leaving. Finally, traditional healers need to be involved in future research projects to evaluate the efficacy of collaboration between modern medicine and TM.

## 5. Conclusions

This study showed that cultural aspects influence the HSBs of people in Pemba City, Mozambique; i.e., local residents heavily relied on traditional healing as their primary source of healthcare. They preferred TM due to their local cultural concepts of ill-health causation and an inherent distrust in modern medicine. In particular, their reliance on inherited customs was strongly supported by community pride, which stemmed from their historical background.

Based on these findings, we strongly recommend that health professionals in Mozambique carefully consider the cultural aspects when providing care for their patients. Moreover, TM needs to be recognized as an important element in their country’s health policy. This study concluded that these are essential to the goal of attaining wide healthcare coverage.

## Figures and Tables

**Figure 1 ijerph-19-02462-f001:**
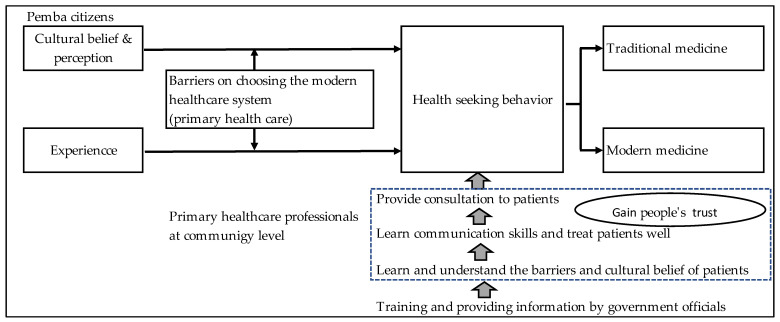
Conceptual framework.

**Figure 2 ijerph-19-02462-f002:**
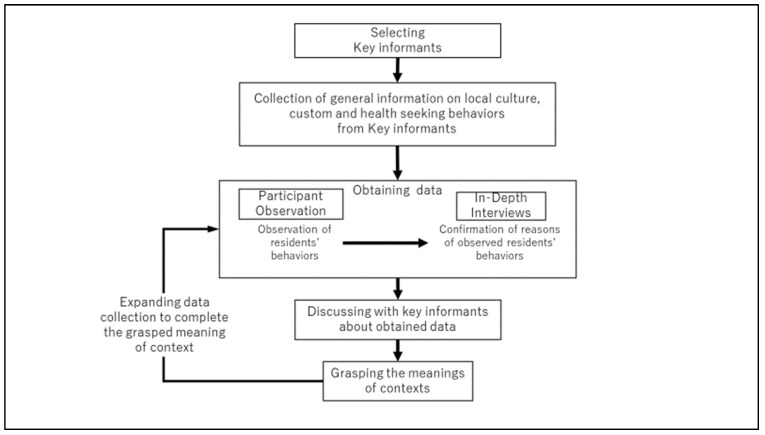
Flowchart for the data collection process.

**Table 1 ijerph-19-02462-t001:** Interview guide.

Variables
• What is “Health” for you? • Have you had any health problems recently? What is it? Why does it disturb you? • Did you go to medical facilities for this problem? • Have you received some methods (treatment) to solve your health problem whether at medical facilities or not? • What did physicians or other medical professionals say about your health problems at hospitals or health centers? (Confirm if there was medical instruction) • Have you followed the medical instruction (such as medication and prevention) by physicians, medical technicians, nurses, pharmacists, or others at hospitals or health centers? • Will you continue to follow the medical instruction? • How long do you intend to continue following this instruction? • Did you understand the medical instruction of physicians or other medical professionals? (Ask concrete questions using their experiences: for example, “Take this medication for reducing body temperature → Did you understand the meaning of reducing the body temperature “you need to take this medication → Did you understand why you need to take the medication?”) • What do you think will happen to you when you don’t follow the instruction? • What do you think will happen to your family and community if you don’t follow the instruction? • Will you go to the medical facilities when a new health problem occurs/a current symptom turns worse? • (Whether yes or no) Why will you go or not go? • Did the medical facilities provide you with what you expected? • What were you told about your health problem from the healer (in case of having consulted with the traditional healer)? • Have you continued to follow the traditional healer’s instructions? • Will you continue to follow the instruction? • How long will you continue to follow the instruction? • Did you understand the traditional healer’s instruction? (Ask concrete questions using their experiences: for example, “Take this medication for reducing body temperature → Did you understand the meaning of reducing the body temperature “you need to take this medication → Did you understand why you need to take the medication?”) • What do you think will happen to you when you don’t follow the instruction? • What do you think will happen to your family and community if you don’t follow the instruction? • Will you go to the healer when a new health problem occurs/a current symptom turns worse? • (Whether yes or no) Why will you go or not go? • Did the traditional healer provide you with what you expected?

**Table 2 ijerph-19-02462-t002:** Socio-demographic characteristics of informants (*n* = 12).

Number	Number
**Age by Decade**	
20 s	2
30 s	4
40 s	2
50 s	4
**Gender**	
Male	8
Female	4
**Tribe**	
Makonde	6
Makuwa	5
Shona	1
**Education**	
No education	1
Under 8th grade	3
Under 12th grade	6
Associate degree	1
Bachelor’s degree	1
**Occupation**	
Teacher	3
Public officer	1
Office worker	2
Health worker	1
Pension recipient	2
Plasterer	1
Guard man	1
No occupation	1

**Table 3 ijerph-19-02462-t003:** Themes, categories, and codes.

Themes	Categories	Codes
Perceptions of health and illness	■ Two types of diseases	• Natural disease • Disease caused by magic
	■ Physical sensation	• Pain • Fever • Five senses
	■ The base of life	• Health for work • Happiness for family
Local perception of causes of diseases	■ Magic of Africa	• Paranormal • Exist only in Africa • Not cured by modern medicine
	■ Diseases of God	• God decides • It’s not magic • Treated in hospital
	■ Indigenous concept	• Invisible force • Magic produced disease
Trust in the indigenous medical system	■ Traditional medicine	• Herbal plants • Oral tradition
	■ Proud to own knowledge	• Have our own medical care • Protected by ancestors • It did not disappear by colonial rule
	■ Curandeirismo	• Intellectual system • African knowledge • Caring for us
Dissatisfaction with governmental medical services	■ Antipathy toward colonial rule	• Hospital treatment is based on colonial knowledge • Our knowledge does not exist in modern medicine • Our knowledge is superior to western medicine
	■ Lack of resource	• Lack of medical staff in hospitals • We cannot rely on modern hospitals
	■ Poor communication with health professionals	• Medical staff treated me badly • Explanation is insufficient

**Table 4 ijerph-19-02462-t004:** The systems of public and traditional medicine.

Category	Public Medical Health System	Traditional Medical Health System
Theory	European Medicine	Curandeirismo
Who provides treatment	Physician and medical technician	Traditional healer (curandeiro)
Medicine	Chemical agents	Medicinal herbs and natural agents (including minerals) Magic
Treatment site	Medical facilities One provincial hospital and nine health centers in each ward	Traditional healer’s house or another site
Access	One provincial hospital and nine health centers, and all facilities are open to anybody who visits	Open access, but the patients need to search by themselves based on personal information
Consultant fee	1 mt (1 United States Dollar = 75 Metical)	Expensive consultant fee (More than 10,000 mt)

## Data Availability

The data presented in this study are available from the corresponding author upon reasonable request. The data are not publicly available because of the need to maintain the participants’ anonymity and data confidentiality.

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
