# Peer review of "Health-Seeking Behaviors in Mozambique: A Mini-Study of Ethnonursing"

_ijerph, 2022, doi:10.3390/ijerph19042462_

Round 1

Reviewer 1 Report

1. The introduction should include a description of the understanding of spirituality which is quite often used in the abstract and text, however in a rather unspefici way. It would be good to take a look at  Peng-Keller, Simon; Winiger, Fabian (2021). Religion and the World Health Organisation: an evolving relationp. BMJ Global Health, 6:e004073 or Peng-Keller, Simon (2019). Genealogies of spirituality. Journal for the Study of Spirituality, 9(2):86-98. (I am not part of the author team). The African traditional medicine has had an important role in introducing the term "spiritual" to health related issues and thus can be supported by the results of your study.

2. The brief ethnohistory should include information on social data regarding religious groups / belonging in the area / city. Since at least two participants are professional "clergy workers" it is of interest how official religion and TM connect or relate.

3. Spelling in figure 1: HSB or HBS? (twice)

4. Study area: can there be a comparison to other African countries regarding density of workforce?

5. The questions of the interview guide is not well explained. How did you get to the questions? Are they based on PO or on literature? The 15th or 16th paragraph suddenly uses the term "healer" or "traditional healer" without asking about him/her before (habe you consulted / would you consult a traditional healer?)

6. Table 1 (interview guide) 5th paragraph: one "s" too much.

7. Table 3 is not well designed. Themes and categories - and the dots in between - are not self-explicatory

8. Line 180: Is there an I missing?

9. Lines 180f: I do not understand this. How can health not be related to illness and vice versa, even according to WHO definition of health? What makes this a "belief"?

10. line 230: Is there no mention of cost dicrepancy?

11: after lines 331f: in my understanding, the study really adds the result how important communication skills are for nurses and physicians if public health care should be more respected. This is important since it does raise the question whether the popularity of TM can be attributed exclusively to cultural/spiritual aspects; obviously the lack in communication skills, espescially listening to and understanding patients' stories and backgrounds, is more important than has been known before. To me, this is one of the really important findings of this study.

Author Response

Responses to Reviewer 1

Thank you for your variable comments and suggestions. We have revised as followed based on your suggestions.

Comment 1:

The introduction should include a description of the understanding of spirituality which is quite often used in the abstract and text, however in a rather unspefici way. It would be good to take a look at Peng-Keller, Simon; Winiger, Fabian (2021). Religion and the World Health Organisation: an evolving relationp. BMJ Global Health, 6:e004073 or Peng-Keller, Simon (2019). Genealogies of spirituality. Journal for the Study of Spirituality, 9(2):86-98. (I am not part of the author team). The African traditional medicine has had an important role in introducing the term "spiritual" to health-related issues and thus can be supported by the results of your study.

(Response)

Thank you for introducing valuable papers to us. Those, especially BMJ, were learning things. The authors have added the following sentences in the Introduction and Discussion sections on Line no. 54-55 and 303-305 accordingly as the reviewer suggested.

It is also indicated that considering the ‘spiritual dimension’ of health is crucial for introducing a new primary health care system in the community.

The World Health Organization (WHO) recognized the importance of working with religious actors on the spiritual domain to deal with primary healthcare for changing the lifestyle and health behavior of the community people.

Comment 2:

The brief ethnohistory should include information on social data regarding religious groups / belonging in the area/city. Since at least two participants are professional "clergy workers" it is of interest how official religion and TM connect or relate.

 (Response)

The authors have added the following sentences from Line no. 79-86 as the reviewer suggested.

However, the main ethnic groups in Mozambique are the Makua, Tsonga, Makonde, Shangaan, Shona, Sena, Ndau, and other indigenous groups. There are approximately 45,000 Europeans, and 15,000 South Asians, constituting less than 2% of the population, as well [22]. The literacy rate in Mozambique is 60.7% (male: 72.6%, female: 50.3%) [23] and more than 46% of people were living below the poverty line of ($0.31 USD) per day in 2018 [24]. Regarding the religious aspect, 26.2% of citizens are Roman Catholic, 18.3% Muslim, 15.1% Zionist Christian, 27% others, and the remaining 13.4% did not list a religious affiliation [25].

Besides, we clarified the word ‘Clerical worker’ with ‘Office worker’ (they were not the clergy workers.). Regarding a question from the reviewer, there is a conflict between official religions such as the Christian church and TM. Usually, traditional healers do not belong to the official religious groups.

Comment 3:

Spelling in figure 1: HSB or HBS? (twice)

(Response)

We revised Figure 1 based on the reviewers’ advice.

Comment 4:

Study area: can there be a comparison to other African countries regarding density of workforce?

(Response)

The authors have added the following sentence from Line no. 139-142 as the reviewer suggested.

On the other hand, the density of doctors in neighboring countries (Tanzania, Malawi, Zambia, Zimbabwe, Eswatini and South Africa) is 0.448 per 1,000 population, and 1.987 nurses and midwives per 1,000 population.

 Comment 5:

How did you get to the questions? Are they based on PO or on literature? The 15th or 16th paragraph suddenly uses the term "healer" or "traditional healer" without asking about him/her before (have you consulted / would you consult a traditional healer?)

 (Response)

The researchers have not consulted with traditional healers. The interview guide was developed based on the participant’s observation (PO). Please see Line no. 176.

Comment 6:

Table 1 (interview guide) 5th paragraph: one "s" too much.

(Response)

The authors have omitted the “s” from the 5th paragraph.

Comment 7:

Table 3 is not well designed. Themes and categories - and the dots in between - are not self-explicatory

 (Response)

To make the table self-explicatory, the authors have revised Table 3 with adding codes.

Comment 8:

Line 180: Is there an I missing?

 (Response)

The authors have added an I on Line no. 225 as the reviewers suggested.

Comment 9:

Lines 180: I do not understand this. How can health not be related to illness and vice versa, even according to WHO definition of health? What makes this a "belief"?

(Response)

To clarify this, the authors have added the explanation on Line no. 226-228 as what the participants’ have mentioned.

Comment 10:

Line 230: Is there no mention of cost discrepancy?

(Response)

Yes, there is a cost discrepancy. The authors have already explained it in table 4 (consultation fee).

Comment 11:

After lines 331: in my understanding, the study really adds the result how important communication skills are for nurses and physicians if public health care should be more respected. This is important since it does raise the question whether the popularity of TM can be attributed exclusively to cultural/spiritual aspects; obviously the lack in communication skills, especially listening to and understanding patients' stories and backgrounds, is more important than has been known before. To me, this is one of the really important findings of this study.

(Response)

Thank you very much for your appreciation.

Reviewer 2 Report

This study incorporates an ethnonursing method to conduct participant observation and in-depth interviews (IDIs) in attempts to investigate health seeking behaviors among Mozambicans living in Pemba City.  Authors assert a need for this study due to the fact that many people in Mozambique die from treatable diseases despite significant advances in healthcare.  Therefore, authors ague that providing culturally competent healthcare services is crucial to improving patient satisfaction and health outcomes in settings where traditional medicine is an integral part of the healthcare system.  The review is as follows:

  1. Lines 35-38 – This sentence should be tightened to make it more compact: “Factors such as fear of mistreatment by care providers and distrust in
    care providers and governments along with financial constraints and ethno-pathogenic perspectives of illnesses have been reported to discourage people from using conventional health services in Mozambique”. For instance, consider ‘ Factors such as fear of mistreatment by and distrust of care providers , wariness of governments, and financial constraints and ethno-pathogenic perspectives of illnesses have been reported to discourage people from using conventional health services in Mozambique’.  Also, it would be helpful to define ‘ethnopathogenic’ for the lay reader.
  2. Lines 45-46 – In “About 80% of the population in Africa rely on traditional medicine”, are there percentages available for Mozambique?
  3. Lines 55-56 – This sentence is unclear - “in a study by Audet et al. involving Mozambicans, people with HIV/AIDS even on the antiretroviral therapy, the majority of them relied…”. Perhaps rephrase this to ‘in a study by Audet et al. that involved Mozambican people with HIV/AIDS who were on antiretroviral therapy, the majority of participants relied…’.
  4. The authors provide an insightful overview of Mozambique and its brief ethnohistory. It would be helpful to hear of the residual effects of colonial occupation.  For instance, are there racial and ethnic health disparities in Mozambique as a result?
  5. Regarding the ethnohistory of people in Mozambique, demographic information would be helpful For instance, what are the racial, ethnic, linguistic, religious, and other characteristics of inhabitants?  These factors can provide insight into health-seeking behaviors.  As authors mention in line 266 that “African countries are multiracial, and therefore contain quite diverse socio-cultural backgrounds”, there should be discussion about the heterogeneity among Mozambican people.
  6. In Figure 1 for the conceptual framework, this framework could be expanded by detailing the mechanisms or pathways through which health professionals understand the influence on HSBs and provide the care considering people’s cultural background.
  7. Line 142 – For the in-depth interviews(IDI), were there incentives provided to participants? It is mentioned that each IDI lasted about 30 to 90 minutes.
  8. Line 174 – The bullets in the table do not line up sequentially.
  9. Line 169 – In ‘3.2. Themes discerned from patterns’, consider making the Themes 1, 2, and 3 to be subthemes.
  10. Line 206 – For ‘Curandeirismo and its healers were called Curandeiros’, these are fascinating concepts. The authors should expand discussion about participants’ preferences for these health systems and the characteristics of these systems that make people prefer them.

Overall, this is an interesting and unique paper on an important topic.  Tending to some clarifying questions may help to improve the paper.

Author Response

Responses to Reviewer 2

Thank you for your variable comments and suggestions. We have revised as followed based on your suggestions.

Comment 1:

Lines 35-38 – This sentence should be tightened to make it more compact: “Factors such as fear of mistreatment by care providers and distrust in care providers and governments along with financial constraints and ethno-pathogenic perspectives of illnesses have been reported to discourage people from using conventional health services in Mozambique”. For instance, consider ‘Factors such as fear of mistreatment by and distrust of care providers, wariness of governments, and financial constraints and ethno-pathogenic perspectives of illnesses have been reported to discourage people from using conventional health services in Mozambique’. Also, it would be helpful to define ‘ethnopathogenic’ for the lay reader.

(Response)

The authors have revised the sentence as the reviewer suggested and define the term ‘ethnopathogenic’ with the disease produced from indigenous causes, not from western medicine but they believe something rooted in ethno belief on Line no. 35-38.

Comment 2:

Lines 45-46 – In “About 80% of the population in Africa rely on traditional medicine”, are there percentages available for Mozambique?

(Response)

The authors have added Mozambicans percentage in the same sentence (Lines no. 48-49) as the reviewer suggested.

Comment 3:

Lines 55-56 – This sentence is unclear - “in a study by Audet et al. involving Mozambicans, people with HIV/AIDS even on the antiretroviral therapy, the majority of them relied…”. Perhaps rephrase this to ‘in a study by Audet et al. that involved Mozambican people with HIV/AIDS who were on antiretroviral therapy, the majority of participants relied…’.

(Response)

The authors have revised the sentence with “in a study by Audet et al. that involved Mozambican people with HIV/AIDS who were on antiretroviral therapy, the majority of participants relied…” on Line 58-60 as the reviewer suggested.

Comment 4:

The authors provide an insightful overview of Mozambique and its brief ethnohistory. It would be helpful to hear of the residual effects of colonial occupation. For instance, are there racial and ethnic health disparities in Mozambique as a result?

(Response)

The authors have added the following sentences on Line no. 79-93.

However, the main ethnic groups in Mozambique are the Makua, Tsonga, Makonde, Shangaan, Shona, Sena, Ndau, and other indigenous groups. There are approximately 45,000 Europeans, and 15,000 South Asians, constituting less than 2% of the population, as well [22]. The literacy rate in Mozambique is 60.7% (male: 72.6%, female: 50.3%) [23] and more than 46% of people were living below the poverty line of ($0.31 USD) per day in 2018 [24]. Regarding the religious aspect, 26.2% of citizens are Roman Catholic, 18.3% Muslim, 15.1% Zionist Christian, 27% others, and the remaining 13.4% did not list a religious affiliation [25]. Mozambique consists of more than 10 tribes. The official language is Portuguese. Religions and languages vary by region and tribe [26].

The health sector in Mozambique has changed throughout history. Before independence, medical facilities only served the Portuguese officers and those who lived in the capital city [27]. After independence and the civil war, the government achieved the reconstruction of the national health system. Now citizens have free access to public medical facilities under the primary healthcare system. However, access to better private facilities is limited to economically wealthy people and foreigners [27].

Comment 5:

Regarding the ethnohistory of people in Mozambique, demographic information would be helpful. For instance, what are the racial, ethnic, linguistic, religious, and other characteristics of inhabitants? These factors can provide insight into health-seeking behaviors. As authors mention in line 266 that “African countries are multiracial, and therefore contain quite diverse socio-cultural backgrounds”, there should be discussion about the heterogeneity among Mozambican people.

(Response)

The authors have added the following sentences regarding heterogeneity among Mozambican people on Line no. 79-93.

Comment 6:

In Figure 1 for the conceptual framework, this framework could be expanded by detailing the mechanisms or pathways through which health professionals understand the influence on HSBs and provide the care considering people’s cultural background.

 (Response)

We revised Figure 1 and added the explanation “Health professionals should understand to provide the care considering the people’s cultural beliefs, better communication to patients, and gain their trust” as the reviewer suggested.

Comment 7:

Line 142 – For the in-depth interviews (IDI), were there incentives provided to participants? It is mentioned that each IDI lasted about 30 to 90 minutes.

(Response)

Yes, an incentive was provided but the authors did not mention it in the manuscript.

Comment 8:

Line 174 – The bullets in the table do not line up sequentially.

(Response)

The authors have lined up the bullets in the table sequentially.

Comment 9:

Line 169 – In ‘3.2. Themes discerned from patterns’, consider making the Themes 1, 2, and 3 to be subthemes.

(Response)

If the authors change them as sub-themes, then there is a need to have a “theme”. But there is no upper-level theme generated. Therefore, the authors have decided to keep it as a theme.

Comment 10:

Line 206 – For ‘Curandeirismo and its healers were called Curandeiros’, these are fascinating concepts. The authors should expand discussion about participants’ preferences for these health systems and the characteristics of these systems that make people prefer them. Overall, this is an interesting and unique paper on an important topic. Tending to some clarifying questions may help to improve the paper.

(Response)

The authors have added the following sentences in the Discussion section from Line no. 290-294 as the reviewer suggested.

Curandeiros were always close to people and their warm attitude earned people’s trust and satisfaction. Visiting curandeiros might be the first step in solving people's health problems. People chose treatment from faraway traditional healers than nearby medical facilities. This indicated people trust their traditional medicine more than modern medicine which was brought by colonial rule.

Reviewer 3 Report

Very interesting topic however the overall presentation should be corrected. I have strong doubt concerning the methodology of the study. The sample is very small (n=12). Thus, it is difficult to speak here about the generalizability of the findings. In addition, the article lacks a solid review of research in the area undertaken. Some research appears in the discussion, but this is insufficient in the reviewer's opinion. Moreover, the theoretical basis of the study should be organized and described in more detail.

Author Response

Responses to Reviewer 3

Thank you for your variable comments.

Comment

Very interesting topic, however the overall presentation should be corrected. I have strong doubt concerning the methodology of the study. The sample is very small (n=12). Thus, it is difficult to speak here about the generalizability of the findings. In addition, the article lacks a solid review of research in the area undertaken. Some research appears in the discussion, but this is insufficient in the reviewer's opinion. Moreover, the theoretical basis of the study should be organized and described in more detail.

(Response)

Thank you for pointing this out. As it is qualitative research, so, we cannot generalize this result. Therefore, we deleted the word “generalized”. Besides, the authors have revised the whole manuscript and added references accordingly as the reviewers suggested.

Round 2

Reviewer 2 Report

The authors have done well to respond to the feedback. The paper is clearer and more comprehensive, including insightful background on heterogeneity among Mozambican people.

Two items for consideration:

  1. In the revised conceptual framework (Figure 1), this sentence is unclear and incomplete – “Health professionals should understand to provide the care considering the people’s cultural beliefs, better communication to patients, and gain their trust…”.  Please review.

  1. Authors mention in their response that an incentive was provided for participants, but that the authors did not mention it in the manuscript. Considering that each in-depth interview (IDI) lasted about 30 to 90 minutes, it is worth mentioning how participants were rewarded for giving their time to be in the study.

Author Response

Reviewer 1

Comments and Suggestions for Authors

The authors have done well to respond to the feedback. The paper is clearer and more comprehensive, including insightful background on heterogeneity among Mozambican people.

Two items for consideration:

1. In the revised conceptual framework (Figure 1), this sentence is unclear and incomplete – “Health professionals should understand to provide the care considering the people’s cultural beliefs, better communication to patients, and gain their trust...”. Please review.

Response

The authors have revised Figure 1 accordingly.

2. Authors mention in their response that an incentive was provided for participants, but that the authors did not mention it in the manuscript. Considering that each in-depth interview (IDI) lasted about 30 to 90 minutes, it is worth mentioning how participants were rewarded for giving their time to be in the study.

 Response

The authors have added the following sentences on the ‘Institutional Review Board Statement’ section on lines no. 379-380.

Reviewer 3 Report

The manuscript has been updated according to Reviewers suggestions. 

Author Response

Reviewer 2

Comments and Suggestions for Authors

The manuscript has been updated according to the Reviewers suggestions.

Response

Thank you very much for your appreciation.